# Surface passivation engineering strategy to fully-inorganic cubic CsPbI$_3$ perovskites for high-performance solar cells

Bo Li[1], Yanan Zhang[1], Lin Fu[1], Tong Yu[1], Shujie Zhou[1], Luyuan Zhang[1] & Longwei Yin[1]

Owing to inevitable thermal/moisture instability for organic–inorganic hybrid perovskites, pure inorganic perovskite cesium lead halides with both inherent stability and prominent photovoltaic performance have become research hotspots as a promising candidate for commercial perovskite solar cells. However, it is still a serious challenge to synthesize desired cubic cesium lead iodides (CsPbI$_3$) with superior photovoltaic performance for its thermo-dynamically metastable characteristics. Herein, polymer poly-vinylpyrrolidone (PVP)-induced surface passivation engineering is reported to synthesize extra-long-term stable cubic CsPbI$_3$. It is revealed that acylamino groups of PVP induce electron cloud density enhancement on the surface of CsPbI$_3$, thus lowering surface energy, conducive to stabilize cubic CsPbI$_3$ even in micrometer scale. The cubic-CsPbI$_3$ PSCs exhibit extra-long carrier diffusion length (over 1.5 μm), highest power conversion efficiency of 10.74% and excellent thermal/moisture stability. This result provides important progress towards understanding of phase stability in realization of large-scale preparations of efficient and stable inorganic PSCs.

[1] Key Laboratory for Liquid-Solid Structural Evolution and Processing of Materials, Ministry of Education, School of Materials Science and Engineering, Shandong University, Jinan 250061, P. R. China. Correspondence and requests for materials should be addressed to L.Y. (email: yinlw@sdu.edu.cn)

Due to suitable direct bandgap, high absorption coefficient, and extra-long carrier diffusion length, excellent optoelectronic property, simple and reproducible solution/vapor-chemistry processing[1–3], organic–inorganic hybrid halide perovskite materials (ABX$_3$, A=CH$_3$NH$_3$, B=Pb, X=Br, I) have been deemed as a promising candidate for light harvester for next-generation high-performance solar cells[4–8]. Despite great progress in photovoltaic performance in the last few years, commercial application of perovskite solar cell (PSC) still suffers from moisture and thermal instability owing to the degradation and volatilization of organic component, which presents the uppermost obstacle in further development and mass production[9]. For this reason, all inorganic halide perovskite formed by substituting the organic cation with cesium (Cs) is an optimal alternative for its native inorganic structure stability, and has demonstrated equally efficient and more stable compared to organic–inorganic halide perovskites[10–13].

Of the various inorganic lead halide perovskites, especially, cesium lead iodide (CsPbI$_3$) in cubic phase (α phase) with a bandgap of around 1.73 eV and a visible-light-absorption spectrum up to 700 nm is the mostly desired light harvester in solar cells[14–16]. However, cubic CsPbI$_3$ can only keep stable at high temperature of above 300 °C[14]. As temperature decreasing to ambient temperature, CsPbI$_3$ suffers from thermodynamically phase transition to undesired orthorhombic phase (δ phase) with a wide bandgap of 2.82 eV (Supplementary Figure 1), exhibiting an extremely poor photovoltaic conversion efficiency (PCE) of only 0.09% in PSC[17]. To overcome this obstacle, composition engineering which pursues a certain amount of bromide (Br) to substitute iodide (I) can be one of efficient methods by balancing the tolerance coefficient between PbX$_6$ octahedron and Cs ions[18–20]. For example, Sutton et al.[18] developed a full set of cesium lead halide films from CsPbBr$_3$ to CsPbI$_3$, demonstrating a stabilized power output of 5.6% and J–V efficiency reaching 9.8% for PSC based on cubic CsPbI$_2$Br, although CsPbI$_2$Br still reverts to δ phase over prolonged exposure in air. Increasing continuously bromide proportion induces more prominent phase stability/moisture-stability, dispiritingly, which brings Br-widened bandgap near or above 2.0 eV compared with the ideal solar spectrum (from 1.1 eV to 1.7 eV)[21]. Another effectual method to stabilize cubic phase CsPbI$_3$ is synthesizing colloidal quantum dots (CQDs) with well-controlled size via hot injection process, and best-performance CsPbI$_3$ solar cells are achieved by assembling cubic phase CsPbI$_3$ CQDs as photoactive layer[22–25]. Unfortunately, the undesired α-to-δ phase transition of Cs-based inorganic perovskite has not been inhibited in the solution-chemistry processed film. It is of great and fundamental challenge to develop effective and facile route to synthesize cubic Cs-based inorganic perovskite film for high-performance PSC for potential large-scale industrial application.

Herein, poly-vinylpyrrolidone (PVP)-induced surface passivation strategy is reported to stabilize inorganic perovskite CsPbI$_3$ with cubic crystal structure via a reproducible solution-chemistry reaction process. The surface chemical state of cubic CsPbI$_3$ crystals synthesized in the presence of PVP is investigated via Fourier transformed infrared (FTIR) and nuclear magnetic resonance (NMR) techniques, demonstrating that decreased surface tension can be conducive to stabilize cubic CsPbI$_3$ even in large scale of film with micrometer scale, due to enhanced electron cloud density on the surface of CsPbI$_3$ originated from chemical bonding between acylamino group in PVP and CsPbI$_3$. The obtained cubic CsPbI$_3$ exhibits extra-long carrier lifetime of 338.7 ns and diffusion length of greater than 1.5 μm, up to an order of magnitude compared to the active layer depth. The fabricated PSCs based cubic CsPbI$_3$ achieves the highest power conversion efficiency of 10.74% and excellent thermal/moisture stability.

## Results

**PVP-induced cubic phase stability studies.** The specific cubic-phase CsPbI$_3$ films were prepared via a simple and reproducible one-pot solution spin-coating process using a mixture of CsI, PbI$_2$, and PVP as a precursor. X-ray diffraction (XRD) patterns of the CsPbI$_3$ films coated on the F-doped SnO$_2$ (FTO) substrates present the difference in the presence and absence of PVP. In the one-pot solution process without PVP, the CsPbI$_3$ film exhibits a prompt transition from cubic phase to orthorhombic phase when prolonging anneal time or cooling to room temperature, as shown in the XRD pattern (black line) in Fig. 1a and the photograph in Fig. 1b. After adding PVP and gradually increasing the concentration to 10 wt%, the CsPbI$_3$ can keep stable cubic phase both at high and room temperature, even stable at exceeding 80 days (Fig. 1a, b; Supplementary Fig. 2). Similarly, the prominent phase stability is demonstrated achievable in full series of inorganic perovskite compositions from CsPbI$_3$ to CsPbBr$_3$ shown in Supplementary Figures 3 and 4.

We further fabricate CsPbI$_3$ film on mesoporous TiO$_2$ via one-step solution spin-coating process with different PVP concentrations. As shown in SEM images of Fig. 1c, d, both orthorhombic and cubic CsPbI$_3$ exhibit high-surface coverage. Compared with irregular grain size distribution of orthorhombic ones, the obtained PVP-induced cubic CsPbI$_3$ film presents a dense grained uniform morphology with comparatively small grain size of around 100 nm. The cross-section morphology of the fabricated solar device architecture is shown in Fig. 1e, consisting mainly of two uniform layers containing a 400 nm mesoporous TiO$_2$/CsPbI$_3$ nanocomposite film and a 100 nm pure CsPbI$_3$ perovskite overlayer. It is shown that the inorganic perovskite materials are fully permeated into TiO$_2$ mesoporous layer, forming a very uniform overlayer with 100% coverage. Intriguingly, tuning anneal time range, the spin-coating obtained CsPbI$_3$ exhibits crystalline size of over 1 μm and high-crystalline with cubic phase structure (Supplementary Figs. 5, 6 and 7), which is different from the previous reports involving of phase transition of perovskite materials in large grain size[22].

In order to gain insight into the PVP stabilization mechanism on cubic CsPbI$_3$, we examine the infrared transmittance spectra of CsPbI$_3$ films (Fig. 2a) for pure PVP, CsPbI$_3$ film synthesized in the presence of PVP, and the CsPbI$_3$ film after removing PVP washed with isopropanol (IPA). The IR spectrum of pure PVP shows absorption bands in the region of 1668, 1421, and 1297 cm$^{-1}$, which are attributed to typically functional groups of C=O, C–H, and C–N stretching vibration in acylamino of PVP, respectively[26,27]. For the IR spectrum of the CsPbI$_3$ film synthesized in the presence of PVP, these characteristic vibrations are still persisted, but only blue-shifting to 1652 cm$^{-1}$ for the C=O stretching, and 1282 cm$^{-1}$ for the C–N stretching, respectively, indicating an interaction effect between functional groups of PVP and precursor ions of CsPbI$_3$. For the CsPbI$_3$ film washed with IPA, characteristic bands for C=O, C–N, and C–H stretching decreases considerably in terms of intensity, while it remains at the same location. A similar binding energy variation of CsPbI$_3$ surface elements can be found in X-ray photoelectron spectroscopy (XPS) measurements in Supplementary Figure 9. The variation tendency demonstrates that PVP is not only absorbed on the surface of CsPbI$_3$ physically, but also functions chemically in formation and stabilization of cubic CsPbI$_3$ through N–C=O bond of acylamino group[27].

On the basis of the above IR information, it is indicated that acylamino group of PVP plays a dominant role on the nucleation and growth of cubic CsPbI$_3$ perovskite film. For further understanding this specific effect of acylamino group of PVP, the liquid-state $^1$H/$^{13}$C NMR measurement is conducted on pure PVP samples and CsPbI$_3$ perovskite synthesized in the presence

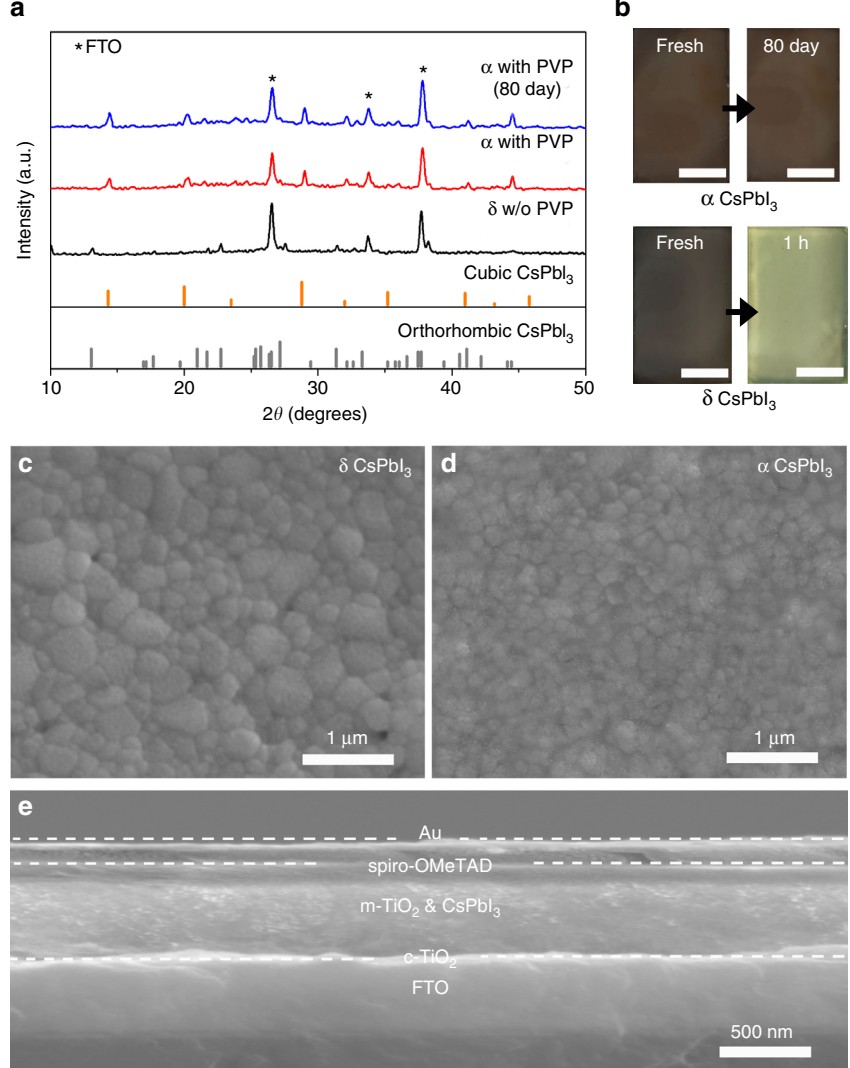

**Fig. 1** Structure and morphology of CsPbI$_3$ films and CsPbI$_3$ perovskite solar cell. **a** X-ray diffraction (XRD) spectra of CsPbI$_3$ with orthorhombic phase (δ, black line), cubic phase (α, red line) and stable cubic phase aging 80 days (blue line). The reference powder pattern for CsPbI$_3$ (cubic and orthorhombic phase) is from Swarnkar et al.[25] **b** Images of prepared orthorhombic and cubic CsPbI$_3$ films aging for different times. Scale bar, 1 cm. **c, d** Scanning electron microscope (SEM) images of the overlayers for orthorhombic and cubic CsPbI$_3$ films deposited on the meso-TiO$_2$ annealing for 5 min at 300 °C. **e** The typical cross-section SEM image of fabricated inorganic perovskite CsPbI$_3$ solar cell

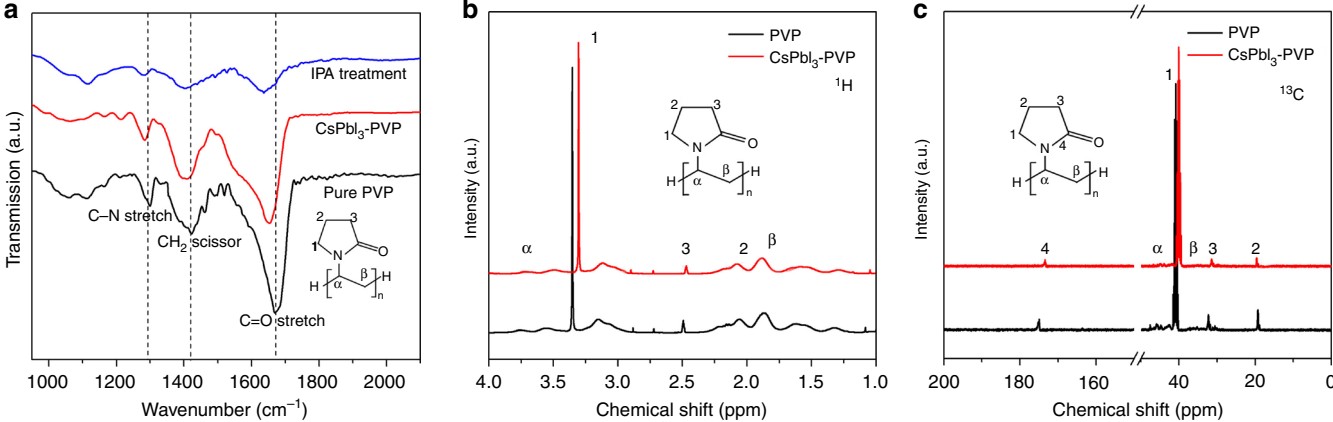

**Fig. 2** Fourier transform infrared and nuclear magnetic resonance spectra of CsPbI$_3$. **a** Fourier transform infrared (FTIR) spectroscopy of pure PVP, cubic phase CsPbI$_3$ films synthesized in the presence of PVP, and cubic CsPbI$_3$ films after IPA treatment. **b, c** $^{1}$H and $^{13}$C liquid-state nuclear magnetic resonance (NMR) spectra of PVP solution and CsPbI$_3$ perovskite solution in the presence of PVP dissolved with DMSO-d$_6$

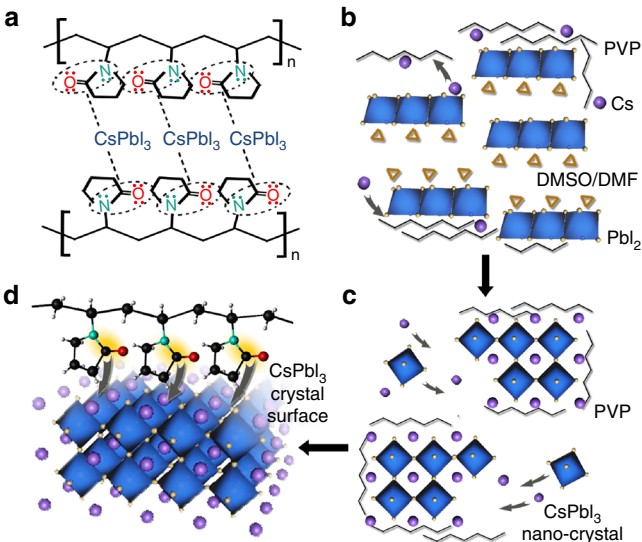

**Fig. 3** Mechanism of PVP-induced cubic phase stability. **a** Schematic diagram of the chemical bonding between $CsPbI_3$ and PVP molecules. PVP molecule contains of long-chain alkyls and acylaminos. The unbounded lone pairs for N/O atoms in acylaminos offer excess electrons and interact with Cs ions in $CsPbI_3$. Mechanism and scheme for the formation of cubic phase $CsPbI_3$ with the assistant of PVP in three stages. **b** $PbI_2$ and Cs ions in DMF/DMSO solvent assemble and interact with PVP molecules spontaneously, and maintain a metastable state. **c** $CsPbI_3$ nanocrystals formed attached on PVP molecules, and remain relatively independent and stable under the effect of PVP molecules. **d** PVP anchored at the surface of $CsPbI_3$ crystals via the combination between N/O and Cs. The negative state in $CsPbI_3$ crystals surface reduces surface tension significantly and stabilizes cubic phase

of PVP in deuterated DMSO-$d_6$ solution, as shown in Fig. 2b, c. In $^1H$ NMR spectra (Fig. 2b) of neat PVP sample, resonance signals attributed to the acylamino group appear at $\delta = 2.5$ and 3.35 ppm, which are characteristic of $CH_2$ attached to C=O group and N atom, respectively[28]. The interaction of unique groups in PVP with precursor ions of $CsPbI_3$ induces a downfield chemical shift of $\Delta\delta \approx 0.5$ ppm for $CH_2$ adjoining with acylamino group. Reversely, almost no variation for the resonances of hydrogen in backbone chain appears. This can be rationally explained in terms of strengthening effects of resonance for organic constituents through the interaction between cesium cations of perovskite and atoms in organic molecules of PVP, reflecting on the shift of chemical resonances[29]. Moreover, the result indicates that the N and O atoms in acylamino group are jointly responsible for the chemical shift and can be as two possible centers for coordination with cesium ions. Furthermore, $^{13}C$ NMR spectroscopy in Fig. 2c show that resonance signal of $\delta = 175$ ppm arising from C=O group undergoes a significant downfield shift of $\Delta\delta \approx 2$ ppm on interaction of PVP with $CsPbI_3$, which is indicative of the coordination-bonding interaction between oxygen atoms of acylamino group and cesium ions in perovskite[30]. In contrast, the resonances at $\delta = 41$ and 43 ppm for $C(1)H_2$ and $C(\alpha)H$ attached to nitrogen atoms exhibit slight chemical shift. Such variation of PVP molecule structure further confirms that there exist two potential centers, i.e. the nitrogen and the carbonyl oxygen, interacting with $Cs^+$ ions of perovskite exposed[31]. Moreover, the oxygen in acylamino group occupies a dominate position in the formation of C=O⋯Cs bonds, since nitrogen in planar conformation of internal amide can only have relatively weak influence.

A conceivable PVP-induced surface tension-driven mechanism for the formation of stable cubic phase $CsPbI_3$ is proposed based on the above experiment facts. It is known that the acylamino group in N-vinylpyrrolidone molecule of PVP has donated lone pairs related to oxygen and nitrogen atoms, which offer a large number of coordination centers. As shown in Fig. 3a, the coordination modus indicates the polymer molecules coordinate onto the surface of $CsPbI_3$ through the oxygen atoms, to a lesser extent, via the nitrogen of N–C=O groups, resulting in a weakening of the C=O bonding and an increasing of N–C bond. At the initial stage (Fig. 3b), PVP molecules initiate to attract cations of $CsPbI_3$ precursors due to long backbone chain and electronegative acylamino group structure. The positive and negative ions of $CsPbI_3$ tend to assemble and bond to form cubic $CsPbI_3$ a metastable state around the N–C=O coordination centers of PVP. With time increasing, more nuclei of $CsPbI_3$ are promptly launched with PVP attached. The PVP molecules, in the meantime, stabilize the $CsPbI_3$ nanocrystals from aggregation owing to the intermolecular rejection effect, as shown in Fig. 3c. For the grown $CsPbI_3$ nanocrystals, long-chain PVP molecules with a large number of acylamino groups anchored at the surface of $CsPbI_3$ provide more coordination polymer units for interactions between oxygen, nitrogen in acylamino and cesium ions of inorganic perovskite. With the growth of $CsPbI_3$ stabilized with PVP, the interactions between N–C=O of acylamino and $Cs^+$ of inorganic perovskite exposed at the surface are enhanced, contributing increasing negative field in $Cs^+$-PVP complex on the surface of $CsPbI_3$ (Fig. 3d), which results in the enhancement of the electron cloud for $Cs^+$ of $CsPbI_3$[26,32]. According to study by Grätzel's group that the surface free energy is a function of the surface tension[33]. While the surface tension is related to charge density[34]. An increase in charge density decreases the surface tension. Therefore, in the $CsPbI_3$-PVP complex, the increase in the electron cloud density may result in low surface tension, thus greatly reduces the surface energy of $CsPbI_3$. As a result, the cubic $CsPbI_3$ can be maintained at ambient temperature. Furthermore, cubic structure of $CsPbI_3$ can even be still maintained after 80 days for the PVP chemically functionalized $CsPbI_3$. Owing to the increase of surface charge originated from the interaction between PVP and $CsPbI_3$, the surface tension of $CsPbI_3$ grains reduced significantly, which plays an essential role in the stabilization of cubic phase $CsPbI_3$.

**Optical and photovoltaic performance**. Weighing the phase stability and power conversion efficiency (Supplementary Figs. 8, 14, 15 and 16), the optimal synthetic condition (10 wt% of PVP, 5 min annealing, and 30 min IPA treatment) was selected and applied for the following optical, electrical and photovoltaic investigation. Figure 4a presents the ultraviolet–visible absorption spectra of the obtained cubic and orthorhombic phase $CsPbI_3$ films. The orthorhombic $CsPbI_3$ exhibits limited visible-light-absorption range less than 450 nm, demonstrating that it is unsatisfactory as an optical active material for solar devices. Oppositely, the cubic $CsPbI_3$ shows a desired absorption width to 700 nm, nearly covering full visible-light region. Furthermore, we investigate the effect of anneal time (crystalline grains) on the optical properties and carry out the PL measurement of cubic $CsPbI_3$ perovskite films. The result shows that, tuning size of cubic $CsPbI_3$ grains, the emission peaks red-shift gradually until to a constant value of around 710 nm (Supplementary Fig. 10).

The time-resolved photoluminescence (TRPL) measurement (Fig. 4b; Supplementary Figure 18) is conducted to investigate the carrier lifetime of cubic and orthorhombic $CsPbI_3$ films. To eliminate the effect of quenching treatment, the $CsPbI_3$ films are deposited on glass slides under the same solution-method and the same thickness. The corresponding steady-state PL spectra of cubic and orthorhombic $CsPbI_3$ films are shown in

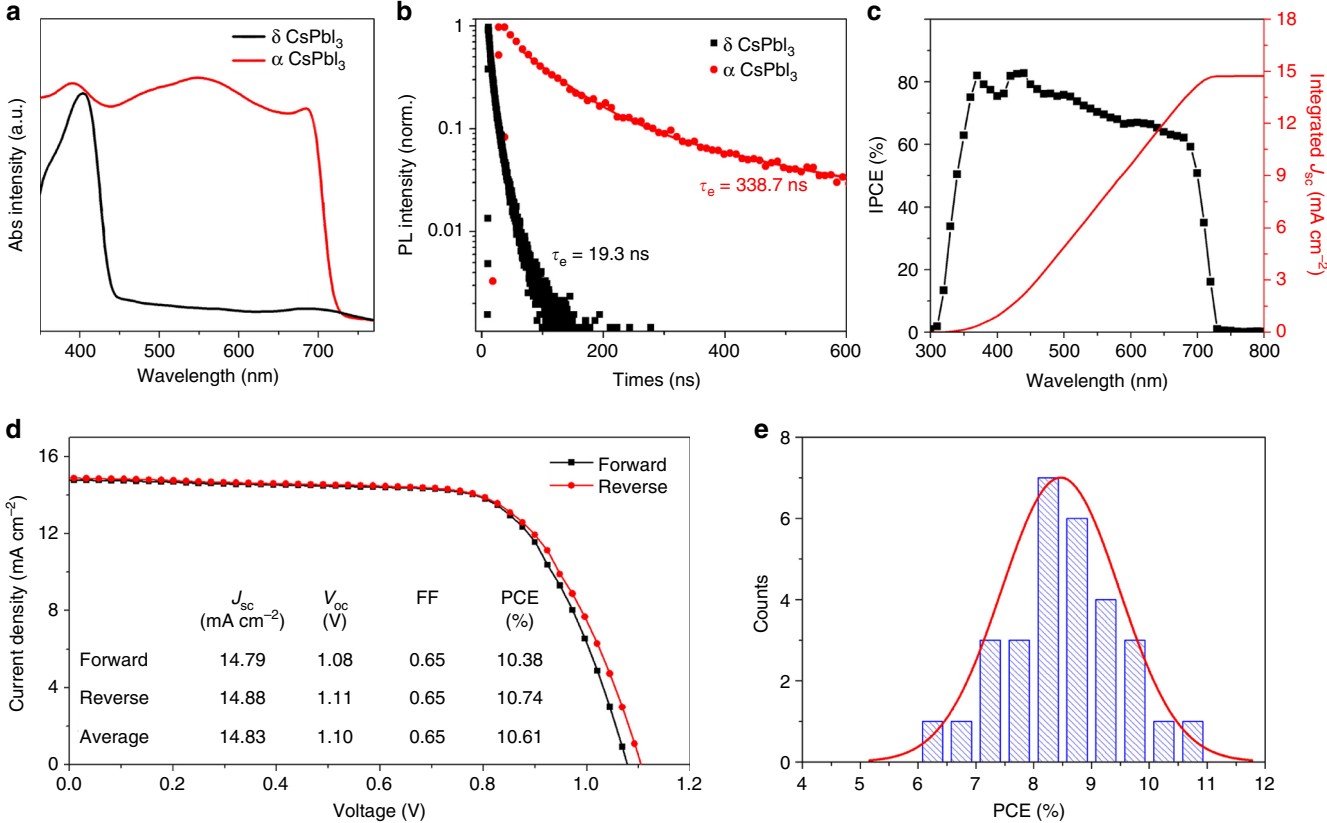

**Fig. 4** Optical and photovoltaic performance of cubic CsPbI$_3$. **a** The ultraviolet–visible (UV) absorption spectra of orthorhombic and cubic CsPbI$_3$ films. **b** Time-resolved photoluminescence (TRPL) spectra of orthorhombic and cubic CsPbI$_3$ films deposited on glass substrates. The excitation wavelength was fixed at 300 nm, the emission wavelengths were set at 412 and 704 nm for orthorhombic and cubic, respectively. **c** The incident photon-to-current efficiency (IPCE) spectrum and corresponding integrated $J_{sc}$ for the best-performance cubic CsPbI$_3$ solar cell. **d** The $J$–$V$ curves for the best cubic CsPbI$_3$ cell measured by forward and reverse scans. The average photovoltaic performance values form the two $J$–$V$ curves are summarized (inset). **e** Histogram of average efficiencies for 30 devices of cubic CsPbI$_3$

Supplementary Figure 11. The PL decay for neat orthorhombic CsPbI$_3$ film exhibits a time-constant of $\tau_e = 19.3$ ns. In contrast, cubic CsPbI$_3$ film shows an extra-long carrier lifetime of $\tau_e = 338.7$ ns. To simulate the carrier diffusion length in perovskite films, only electron/hole extraction layers and inorganic perovskite layer (i.e., TiO$_2$/CsPbI$_3$ and CsPbI$_3$/spiro-OMeTAD) are fabricated via same solution-chemistry processing and the same thickness with the fabricated cell, the PL decay curves with electron/hole extraction layers are shown in Supplementary Figures 12 and 13, the PL decay dynamics are modeled via accounting the excitations number and distributions based on the one-dimensional diffusion equation[1].

$$\frac{\partial n(x,t)}{\partial t} = D\frac{\partial^2 n(x,t)}{\partial x^2} - k(t)n(x,t) \qquad (1)$$

in which $n(x,t)$ is the number of excitations within a certain thickness of perovskite film, $k(t)$ is the PL decay rate without quenching layer, and $D$ is the diffusion coefficient. Table 1 shows the carrier diffusion length for both orthorhombic and cubic CsPbI$_3$ films, which depends on electron or hole quenching layer used, and it is assumed that all photogenerated carriers reach the quenching interface. It is clear the diffusion length of both electron and hole in orthorhombic CsPbI$_3$ film is around 120 nm. However, for cubic CsPbI$_3$ film, the carriers exhibit a diffusion length for electrons over 1 μm, and even over 1.5 μm. As reported (Supplementary Table 1), the average carrier diffusion length in organic–inorganic hybrid perovskites MAPbI$_3$ and FAPbI$_3$ is 129

**Table 1 The carrier diffusion constant (D) and diffusion length (L$_D$) simulated form PL decays using the diffusion model**

| Phase | Species | D (cm$^2$ s$^{-1}$) | L$_D$ (nm) |
|---|---|---|---|
| Cubic | Electrons | 0.061 ± 0.016 | 1566 ± 254 |
| | Holes | 0.057 ± 0.013 | 1427 ± 238 |
| Orthorhombic | Electrons | 0.014 ± 0.009 | 121 ± 51 |
| | Holes | 0.011 ± 0.007 | 117 ± 35 |

The errors arise predominantly from perovskite film thickness variations, which is ±50 nm for both orthorhombic and cubic CsPbI$_3$ films

and 813 nm, respectively[1,8]. In addition, in pure/mixed Br based inorganic perovskite, the carrier diffusion length is less than 200 nm[12]. The ultra-long carrier diffusion length not only originates from the excellent carrier transport capability of cubic CsPbI$_3$, but also from the inhibition of defect recombination via the surface passivation effect, which provides the feasibility in planar-structure PSCs or even thicker light-absorption layers.

On the basis of optical and carrier transport properties, we conduct the photovoltaic measurements of the cubic CsPbI$_3$ PSCs fabricated with mesoporous TiO$_2$ scaffold. Of the solar devices acquired, Fig. 4d depicts the current–voltage ($J$–$V$) curves measured via forward and reverse bias sweep for the best-performance PSCs. The corresponding photovoltaic parameters under the optimized conditions with an active area of 0.09 cm$^2$, including of short-circuit current density ($J_{sc}$), open-circuit

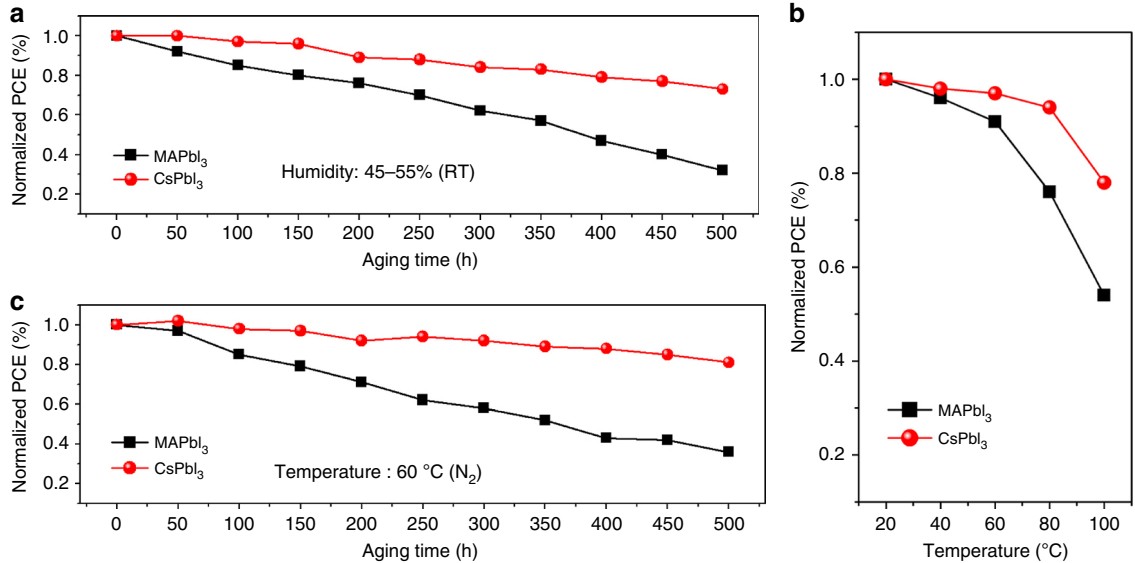

**Fig. 5** Moisture and thermal stability investigation of perovskite solar cells based cubic CsPbI₃. **a** Efficiency evolution of the devices exposed in ambient air under relative humidity of 45–55% without any sealing. The measurements were carried every 50 h during 500 h. **b** Efficiency variation as a function of temperature from 20 to 100 °C. The PCEs were measured under nitrogen atmosphere after an equilibration time of 30 min at each temperature setting. **c** Efficiency evolution of the cells in a nitrogen atmosphere at 60 °C during 500 h

voltage ($V_{oc}$), fill factor (FF), and PCE values are summarized in the insert of Fig. 4d. The $J_{sc}$, $V_{oc}$, and FF for forward sweep of the device are 14.79 mA cm$^{-2}$, 1.08 V, and 65%, respectively, corresponding to a PCE of 10.38% under standard AM 1.5 G condition. With faint hysteresis, the solar device for reverse sweep exhibits a $J_{sc}$ of 14.88 mA cm$^{-2}$, a $V_{oc}$ of 1.11 V and a PCE of 10.74%, which are higher than those of previous reports on CsPbBr₃ and CsPbBr₃₋ₓIₓ (Supplementary Table 2)[18,19,35]. Moreover, the stability of $J_{sc}$ and PCE for both devices is shown in Supplementary Figure 19. The cubic CsPbI₃ device shows a stable output with a $J_{sc}$ of 13.1 mA cm$^{-2}$ and a PCE of 10.0%. The corresponding incident photo-to-current efficiency (IPCE) spectrum in Fig. 4c for the best cell exhibits a broad plateau of over 60% between 350 and 700 nm. The integrated $J_{sc}$ of 14.7 mA cm$^{-2}$ is in good agreement with the current density acquired from the current–voltage measurement. Intriguingly, compared with other inorganic PSCs[18,19,35], the CsPbI₃ based solar cell exhibits a much higher $J_{sc}$, which can be attributed to the extended visible-light-absorption range and extra-long carrier diffusion length for CsPbI₃, beneficial to more photoelectrons/hole generations and captures by corresponding transport layers. Moreover, the PVP covered on CsPbI₃ grains decreases surface defects and suppresses nonradiative recombination, significantly (Supplementary Figure 17). Figure 4e shows a histogram of average PCEs from all of the cubic CsPbI₃ PSCs fabricated under the same condition for the repeatability purpose. Over 70% of the devices exhibit over 8% PCE, and the average PCE summarized shows 8.50%, which is better than most current stable and efficient CsPbI₂Br perovskite cells (Supplementary Figure 20).

**Moisture and thermal stability**. The excellent stability of PSCs is an essential factor for the reproducibility and commercial application. To investigate the moisture and thermal stability under different conditions, the performance of inorganic cubic perovskite (CsPbI₃) PSCs with average PCE is comparatively measured with that of solar cells based on typical organic–inorganic hybrid perovskite (MAPbI₃). The ambient-humidity-stability test was conducted under ambient condition for 500 h without encapsulation (average humidity of 45–55% with temperature of

26 °C fixed). Fig. 5a shows the device moisture-stability as a function of aging time in terms of normalized power conversion efficiency (PCE). During 500 h, the cell of MAPbI₃ shows a dramatic drop with 70% efficiency loss with respect to the fresh solar cell. Comparatively, cubic CsPbI₃ based device exhibits a better moisture-stability with 75% retention after 500 h. Figure 5b shows the thermal-stability measurement of PSCs with cubic CsPbI₃ and MAPbI₃, which was conducted at different temperature ranging from 20 to 100 °C. It is clear that, with the increasing of temperature, the inorganic cubic perovskite exhibits more prominent thermal stability, showing over 90% efficiency retention even at 80 °C. It is worth noting that, as increasing the temperature to 100 °C, the devices of both CsPbI₃ and MAPbI₃ show obvious decay in PCE, which might result from the failure of the organic hole transport material. For further investigating the long-term thermal stability of the inorganic perovskite CsPbI₃ solar cell, we measured the device performance as stored at high temperature (60 °C) under normal sunlight exposure, which is shown in Fig. 5c. During 500 h measurement, the cubic CsPbI₃ based PSC shows a slight efficiency decay of only around 10%, demonstrating an outstanding superiority in thermal stability compared to MAPbI₃ based solar cell (70% efficiency loss). Notably, the thermal-stability efficiency test for inorganic perovskite exhibits a slower decay rate than the humidity stability. The result demonstrates that the CsPbI₃ inorganic perovskite possesses more outstanding stability, especially in thermal stability.

## Discussion
In summary, we developed a surface passivation engineering for preparing long-term stable cubic phase CsPbI₃ films via a reproducible solution-chemistry process with the assistant of PVP. We proposed a plausible mechanism for the formation of stable cubic CsPbI₃ by investigating the surface chemical states of the perovskite crystals. The decreased surface tension can be obtained to stabilize CsPbI₃ grains in cubic phase even in micrometer scale, due to electron cloud density enhancement on the surface of CsPbI₃ originated from chemical bonding between acylamino in PVP and CsPbI₃. Furthermore, we found the

obtained cubic CsPbI$_3$ exhibits prominent photoelectronic properties with extra-long carrier lifetime of 338.7 ns and diffusion length of greater than 1.5 μm, up to an order of magnitude compared to the absorption depth. Based on this strategy, we have achieved the highest PCE of 10.74%, as well as excellent thermal/moisture stability in the fabricated inorganic PSCs. This result provides important progress towards the understanding of phase stability in the realization of large-scale preparations of efficient and stable inorganic PSCs.

## Methods

**Inorganic perovskite film preparation**. A certain amount of PVP (Aladdin, K13-18) was first dissolved in mixed solvent with DMF (Aladdin, 99.9%) and DMSO (Aladdin, anhydrous) (1:1) and stirred for 30 min at room temperature. The CsPbI$_3$ precursor solution (0.8 M) was synthesized by dissolving stoichiometric CsI (Aladdin, 99.9%) and PbI$_2$ (Aladdin, 99.9%) in above solution, then, was stirred at 90 °C for 1 h on hot plate. The perovskite precursor solution was spin-coated on glass substrate or mesoporous TiO$_2$ film at 2500 rpm for 45 s and sintered at 300 °C for 5 min to form CsPbI$_3$ film. The other series of cesium lead halide perovskite films with different iodide/bromide proportions were synthesized by changing the percentage of PbI$_2$ and PbBr$_2$, and keeping other methods fixed.

**Device fabrication**. A compact TiO$_2$ layer (about 50 nm) was deposited on the FTO substrates (OPV Tech) which were ultrasonically washed and underwent oxygen plasma treatment by spin-coating a mildly acidic solution of titanium isopropoxide in ethanol (2000 rpm, 30 s), and annealed at 500 °C for 30 min. A 400–500 nm thick mesoporous TiO$_2$ film was coated on the compact layer via TiO$_2$ paste (2500 rpm, 45 s), next, heated for 30 min at 500 °C. Then, the CsPbI$_3$ active layer was deposited on mesoporous TiO$_2$ scaffold with 300 °C annealing treatment for 5 min. To increase the surface coverage of the inorganic perovskite, the substrates with TiO$_2$ coated were preheated at 150 °C. After cooling to room temperature, the substrates were immersed into IPA for 30 min to remove redundant PVP. Finally, an around 200 nm hole transport layer of spiro-OMeTAD (OPV Tech, 99.5%) was spin-coated at 2500 rpm for 30 s and a 50 nm gold counter electrode was prepared by thermal evaporation. The optical active layer and hole transport layer were fabricated in glove box.

**Characterization**. The XRD spectra of inorganic perovskite films were measured by Phillips Rigaku D/Max-kA X-ray diffractometer. The surface-section/cross-section morphologies of the perovskite films were characterized using field-emission scanning electron microscopy (FESEM, SU-70). The high-resolution transition electron microscope (HRTEM, Phillips, Tecnai 20U-Twin) was applied to analyze the structures and morphologies of the perovskite crystalline grains. The ultraviolet–visible absorption spectra of the perovskite films were recorded on the TU-1901 spectro-photometer. The FTIR spectra (NEXUS 670) were used to measure the surface functional group of the films. The liquid-state NMR were conducted by VNMRS 600. The XPS measurement was carried out using the Escalab 250Xi electron spectrometer via monochromatized Al Kα radiation ($hv =$ 1486.7 eV). The TRPL measurement were carried by FLS920 all functional fluorescence spectrometer (Edinburgh) with an excitation wavelength of 400 nm. The photovoltaic performances ($J$–$V$ curves) were analyzed by a solar simulator (Newport, Class 3 A, 94023 A) set an AM 1.5 G simulated sunlight (100 mW cm$^{-2}$) equipped with a Keithley 2420, and the solar cells were measured using a metal aperture to define the active area to be around 0.09 cm$^2$. The IPCE was characterized using a power source (Newport 300 W xenon lamp, 66920) equipped with a monochromator (Newport Cornerstone 260) and a multimeter (Keithley 2400) at 100 mW cm$^{-2}$, AM 1.5 G illumination, and was corrected by a calibrated Si-reference cell (NREL).

**Data availability**. All data used in this study are available from the corresponding authors upon reasonable request.

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

## Acknowledgements

We acknowledge support from the project supported by the State Key Program of National Natural Science of China (No.: 51532005), the National Nature Science Foundation of China (No.: 51472148, 51272137), and the Tai Shan Scholar Foundation of Shandong Province.

## Author contributions

L.Y. initiated and directed the study. B.L. conceived the original research idea. B.L. and Y.Z. conducted most of the device fabrication and measurements. S.Z. contributed to the deposition of electron extraction layer. T.Y. contributed to the deposition of hole extraction layer. L.F. contributed to the structural characteristics. L.Z. provided the mechanism idea. The manuscript was co-written by L.Y. and B.L. All authors contributed to the discussion and revising of this manuscript.

## Additional information

**Competing interests:** The authors declare no competing interests.

