## [Peer Review File · Nature Communications]

Reviewers' comments:

Reviewer #1 (Remarks to the Author):

Bo Li et al. reported the surface passivation strategy to stabilize the fully inorganic cubic phase CsPbI₃ perovskite for solar cells using poly-vinylpyrrolidone (PVP). The mechanism behind the stabilization by the PVP was also explored using FTIR, solution ¹H and ¹³C NMR spectroscopy, XRD, and XPS. The device obtained from this strategy demonstrated highest performance for the CsPbI₃ based perovskite solar cells.

Device properties were characterized using XRD, UV-vis, steady-state PL, TRPL, and SEM. The conclusions are reasonable and supported by the experimental evidences.

Nevertheless, some explanations, especially the discussions related to the Fig.3, contains many misleading statements.

Following points must be fully addressed in the revised manuscript.

1. In Fig.3d and TOC, there is negative charges on the PVP.

PVP is a neutral molecule. There may be delocalized electrons along the O=C-N backbone, however, it is not negative charge.

What does "electronic aggregation and electronic cloud field enhancement" means in the abstract?

There are no such aggregations in the proposed system.

Line 243: "excess electronic aggregation on the surface of ..." Again, there is no such aggregation.

Electrons may be delocalized over the individual O=C-N backbones, but that is it. They are localized on each pyrrolidone groups. In fact, no evidence was shown that electrons go over many pyrrolidone groups in polymer chains in this manuscript.

I think there is also some confusion about the zwitterion chemistry in the O=C-N backbone, and all the related discussions should be revised.

2. In the abstract, "fabtunderstanding..." ?

3. Line 95: it goes "... 400 nm mesoporous TiO₂/CsPbI₃ nanocomposite film and a 100 nm pure CsPbI₃ perovskite layer ..." None of these statement can be confirmed in the SEM image in the Fig.1e. Need better images.

4. In the experimental section, Lines: 263-265, description about annealing treatment at 300 degree-C is completely missing.

5. The degree-C symbol is missing in the manuscript.

Reviewer #2 (Remarks to the Author):

This manuscript by Li et al. reports a synthesis route to obtain phase-stable CsPbI₃ perovskite films, and use these as absorber layers in photovoltaic devices. They find that by adding poly-vinylpyrrolidone (PVP) to the precursor solution, CsPbI₃ can be stabilized in its black perovskite phase instead of undergoing a phase transition to an undesired yellow crystal structure. Using FTIR and NMR measurements, the authors show that PVP absorbs on the CsPbI₃ surface and propose that its phase-stabilization is related to a reduction in the surface tension. Finally, they show that these cubic

CsPbI₃-based devices have improved thermal stability compared to devices based on the organic-inorganic methylammonium lead iodide (MAPbI₃).

In view of its superior thermal stability and bandgap value, cubic CsPbI₃ is a suitable candidate for tandem solar cells. However, it is challenging to maintain polycrystalline CsPbI₃ in its black phase under ambient conditions and thus, the development of phase-stable cubic CsPbI₃ is an important step toward commercialization of perovskite-based solar cells.

I think this manuscript will be of interest to the readership of Nature Communications and therefore recommend publication after the authors address the following issues:

1. Page 2, line 60: "another effectual method to stabilize cubic CsPbI₃ is decreasing grain size" actually suggests that increasing the surface energy (i.e. decreasing grain size) is favorable for phase-stabilization of cubic CsPbI₃. This seems in contrast with the proposed mechanism of PVP stabilizing cubic CsPbI₃ by "effectively decreasing surface tension and thus surface energy" (page 1, line 19).
2. On page 2, line 64, the authors state that "Unfortunately, the undesired α -to- δ phase transition of Cs-based inorganic perovskite has not been inhibited in the solution-chemistry processed film." I think that the authors should instead acknowledge previous related work on phase-stabilization of Cs-based inorganic perovskites, such as the recent papers from Zhang et al. (Sci. Adv. 2017, DOI: 10.1126/sciadv.1700841) and Hu et al. (ACS Energy Lett., 2017, 2, pp 2219–2227).
3. On page 6, line 173 and 174, 'time-constants' are given for the PL lifetimes. Please explain how are these derived from the data.
4. Details on how the diffusion lengths are determined (i.e. what was used for $k(t)$ and $n(x,t)$ etc.) are missing. Also, were the samples excited through the extraction layer or from the other side, what was the laser intensity and how did the authors determine the optical attenuation?
5. On page 7, line 189 and page 8, line 209/210, the authors claim that PVP inhibits defect recombination, which is highly speculative based on the data shown. In order to provide evidence for this, the authors could for instance compare to cubic CsPbI₃ without PVP, which can be temporarily maintained in the black phase by heating to > 300 C followed by rapid cooling (Ref. 14 in manuscript).
6. In figure 4b), the initial shape of the red TRPL decay looks a bit odd, like it's off-scale.

Some minor issues:

1. The term 'stability' is used in different contexts. It should therefore be clear to the reader whether the authors refer to crystal phase stability or thermal/moisture stability. Also, although a crystal phase can be metastable, it does not make sense to refer to a phase transition as metastable (p2 line 51).
2. Page 6, line 161, please specify the wt-% of PVP used for the optical, electrical and photovoltaic characterization.
3. Figure captions S6 and S7 "PVP proportion fixed". Please specify the concentration.
4. Page 7, line 207, 'compared with other inorganic perovskite solar cells': references are missing.

Reviewer #3 (Remarks to the Author):

In the manuscript "Surface Passivation Engineering Strategy to Fully-Inorganic Cubic CsPbI₃ Perovskites for High-Performance Solar Cells" by Longwei Yin et al, the authors reported the stabilization of desired cubic phase of CsPbI₃ for photovoltaic. Considering the important and interesting results reported, one would like to recommend the manuscript to be accepted after revising several issues.

Issues:

1. Many typos in the manuscript (for examples, "Sate" Key Program of National Natural Science). The authors should check the manuscript thoroughly.
2. Comparison of current state-of-the-art Cs-based photovoltaics (both CsPbI₃ and CsPbIBr) should be put in a table.
3. 65% FF is still lower than organometallic lead halide perovskites, any suggestion for further improvement?
4. The last and most importantly, the authors showed that PVP stabilized CsPbI_xBr_{3-x} film photographs and UV-vis absorption spectra but no solar cells performance (only CsPbI₃ ones). Authors should at least fabricate CsPbI₂Br devices with reported PVP additive and show their performance since, to date, highest reported Cs-based inorganic perovskite solar cell efficiency (PCE = 11.8%) is from the CsPbI₂Br active layer.

Response to Reviews

Dear Editor and referees,

Thank you very much for your prompt and patient treatment of this manuscript. We would like to express our heartfelt gratitude and appreciation to you and the reviewers for your critical comments and constructive suggestions. We have tried our best to revise our manuscript according to the comments provided by the editor and the three reviewers. We are sending the revised manuscript for your kind considerations. The revisions in detail are as follows.

Response to Reviewer #1

1. In Fig.3d and TOC, there is negative charges on the PVP.

PVP is a neutral molecule. There may be delocalized electrons along the O=C-N backbone, however, it is not negative charge.

(1) Answer and Modification:

The symbol of “negative charge” in Fig. 3d and TOC represents donated lone pairs from oxygen and nitrogen atoms. For avoiding misunderstanding, the symbols have been modified as shown in revised Fig. 3 and TOC.

What does “electronic aggregation and electronic cloud field enhancement” means in the abstract? There are no such aggregations in the proposed system.

(2) Answer and Modification:

It is known that the acylamino group in PVP molecule has donated lone pairs related to oxygen and nitrogen atoms, which offer a large number of coordination centers. In the present work, FTIR and NMR experimental results demonstrate that a coordination interaction takes place between cesium atoms of CsPbI₃ and oxygen/nitrogen atoms from PVP molecular anchored onto the surface of CsPbI₃. Therefore, compared to pure CsPbI₃ without PVP, the PVP passivated CsPbI₃ exhibits enhanced electron cloud density of CsPbI₃ induced by acylamino group from PVP^{R1-R3}.

For more rigorous description, we revise the sentence “electronic aggregation and electronic cloud field enhancement” to “It is revealed that interaction between acylamino groups of PVP and Cs⁺ ions of CsPbI₃ induces electron cloud density enhancement on the surface of CsPbI₃, effectively decreasing surface tension and thus lowering surface energy, which can be conducive to stabilize cubic structure of CsPbI₃ even in micrometer scale.” **The revision is also shown in the Line 7-9 in abstract of manuscript (red font with underline)**

Line 243: “excess electronic aggregation on the surface of ...” Again, there is no such aggregation. Electrons may be delocalized over the individual O=C-N backbones, but that is it. They are localized on each pyrrolidone groups. In fact, no evidence was shown that electrons go over many pyrrolidone groups in polymer chains in this manuscript.

(3) Answer and Modification:

The proposition of “excess electronic aggregation on the surface of ...” is intended to explain the enhancement of negative surface field induced by donated pairs from acylamino group, which

might be ambiguous and may cause misunderstanding. The “excess electronic aggregation” in Line 243 has been removed in the revised manuscript. The whole sentence has been revised as “The decreased surface tension can be obtained to stabilize CsPbI₃ in cubic phase even in micrometer scale, due to electron cloud density enhancement on the surface of CsPbI₃ originated from chemical interaction between acylamino in PVP and CsPbI₃.” The revision is also shown in Line 4-6 of Conclusions Section in manuscript (red font with underline).

I think there is also some confusion about the zwitterion chemistry in the O=C-N backbone, and all the related discussions should be revised.

(4) Modification: According to the suggestions, the related discussions have been revised as following.

Line 5 Paragraph 3 in Introduction Section: “due to enhanced electron cloud field on the surface of CsPbI₃ originated from chemical interaction between acylamino group in PVP and CsPbI₃.”

Line 14 Paragraph 5 in Result and discussion Section: “With the growth of CsPbI₃ stabilized with PVP, the interaction between N—C=O of acylamino and Cs⁺ of iCsPbI₃ is enhanced, contributing to increased negative field in Cs⁺-PVP complex on the surface of CsPbI₃ (Fig. 3d), which results in the enhancement of the electron cloud for Cs⁺ of CsPbI₃.”

Line 19 Paragraph 5 in Result and discussion Section: “Therefore, in the CsPbI₃-PVP complex, the increase in the electron cloud density may result in low surface tension,”

The revisions are all shown in the revised manuscript (red font with underline).

2. In the abstract, “fabt understanding...” ?

Answer and Modification: The “fabt understanding...” might be caused due to file conversion and font format errors. A revised edition has been provided in Line 13 in the revised manuscript.

3. Line 95: it goes “... 400 nm mesoporous TiO₂/CsPbI₃ nanocomposite film and a 100 nm pure CsPbI₃ perovskite layer ...” None of these statement can be confirmed in the SEM image in the Fig.1e. Need better images.

Answer and Modification: According to the suggestion, the better cross-section SEM image of fabricated CsPbI₃ perovskite solar cell has been supplemented and substituted for Original image as shown in Fig. 1e.

4. In the experimental section, Lines: 263-265, description about annealing treatment at 300 degree-C is completely missing.

Modification: The description about the annealing treatment has been revised and supplemented in Line 5, Paragraph 2 of Experimental Section according to the suggestion.

5. The degree-C symbol is missing in the manuscript.

Modification: The missing of degree-C symbol might be owing to the type of font. The degree-C symbols have been corrected and supplemented in all manuscript.

Response to Reviewer #2

1. Page 2, line 60: “another effectual method to stabilize cubic CsPbI₃ is decreasing grain size” actually suggests that increasing the surface energy (i.e. decreasing grain size) is favorable for phase-stabilization of cubic CsPbI₃. This seems in contrast with the proposed mechanism of PVP stabilizing cubic CsPbI₃ by “effectively decreasing surface tension and thus surface energy” (page 1, line 19).

Answer and modification:

It is known that Gibbs free energy is mainly determined by surface area except for volume free energy. In the present work, during the formation process of perovskite phase, Gibbs free energy is predominantly affected by surface tension^{R4}. Decreasing grain size can increase surface energy, which is opposite to the minimum steady state of energy. In fact, all of the cubic-phase-stable CsPbI₃ acquired by “decreasing grain size” are CsPbI₃ quantum dots. It should be pointed that for the structure stability of quantum-dot-sized CsPbI₃ (i.e. high surface energy), the long-chain functional groups surrounding CsPbI₃ quantum may play an critical role on the cubic phase stability, which is the essence of the presence of cubic phased quantum dots at room temperature and no agglomeration.

The mechanism “effectively decreasing surface tension and thus surface energy” proposed in the present work coincides with the classical Gibbs free energy principle. Specifically, we find that CsPbI₃ can keep cubic phase stability not only in nanometer scale but also in micrometer scale in the presence of PVP, as shown in Fig. S6 and S7. Therefore, we propose “effectively decreasing surface tension and thus surface energy”.

Therefore, to avoid misunderstanding, the related expression is revised as following.

“Another effectual method to stabilize cubic CsPbI₃ is decreasing grain size...” is revised as “Another effectual method to stabilize cubic phase CsPbI₃ is synthesizing colloidal quantum dots (CQDs) with well-controlled size via hot injection process, and best-performance CsPbI₃ solar cells are achieved by assembling cubic phase CsPbI₃ CQDs as photoactive layer.” as shown in **Line 13-15, Paragraph 2 of Introduction Section.**

2. On page 2, line 64, the authors state that “Unfortunately, the undesired α -to- δ phase transition of Cs-based inorganic perovskite has not been inhibited in the solution-chemistry processed film.” I think that the authors should instead acknowledge previous related work on phase-stabilization of Cs-based inorganic perovskites, such as the recent papers from Zhang et al. (Sci. Adv. 2017, DOI: 10.1126/sciadv.1700841) and Hu et al. (ACS Energy Lett., 2017, 2, pp 2219–2227).

Answer: These two related works on phase-stabilization of CsPbI₃ mentioned by referee all demonstrate the improvement of cubic-phase-stability and all have vital significance in the development of inorganic perovskite materials. We supplemented these two works in the revised manuscript as **reference 15 and 16 in Line 3, Paragraph 2 of Introduction Section.** However, in terms of method, they all focus on the composition engineering, which is analogous with the previous works involving halide-site substitution by pursuing a certain amount of bromide (Br) to

substitute iodide (I) as an efficient method to balance the tolerance coefficient between PbX_6 octahedron and Cs ions. The difference is that, Zhang et al. introduce EDA cation into A-site to fabricate double-phase structure, and Hu et al. incorporate bismuth cation in to B-site to substitute lead with larger atomic radius. Actually, they did not achieve a real sense of cubic-phase-stability for “pure $CsPbI_3$ ”. Comparatively, in this work, we propose a brand-new poly-vinylpyrrolidone (PVP) induced surface passivation engineering strategy, achieving extra-long-term-stable cubic perovskite $CsPbI_3$ film via a reproducible one-pot solution-chemistry spin-coating process. It is revealed that decreasing surface tension is essential to stabilize cubic-phase structure of $CsPbI_3$, providing more insight in understanding phase stability of cubic perovskite $CsPbI_3$ film.

3. On page 6, line 173 and 174, ‘time-constants’ are given for the PL lifetimes. Please explain how are these derived from the data.

Answer:

The ‘time-constants’ are acquired by fitting the PL decay curve using fitting software F900 installed in the time-resolved photoluminescence measurement system FLS920 functional fluorescence spectrometer. Specifically, parameters describing the photoluminescence dynamics in the absence of any quencher are required inputs in the diffusion model. These are obtained by fitting the background-corrected PL measured from perovskite films with a stretched exponential decay function^{R5,R6}. And errors in the fitting parameters were determined by examining the reduced surfaces obtained by independently varying each fitting parameter.

4. Details on how the diffusion lengths are determined (i.e. what was used for $k(t)$ and $n(x,t)$ etc.) are missing. Also, were the samples excited through the extraction layer or from the other side, what was the laser intensity and how did the authors determine the optical attenuation?

Answer and Modification:

The diffusion lengths are determined according 1D diffusion equation and related derivation by referring to the reports from Stranks et al. and Shaw et al.^{S1, S2}. The values of excitation concentration, decay rate and diffusion coefficient are all acquired via Fitting PL decay curves and calculated from individual points using SPSS data statistics software. For the missing details and questions mentioned, more detailed explanations involving diffusion length statistics, equipment parameters and operation are supplemented in supplement information.

The supplemental contents are as follows:

To simulate the carrier diffusion length in perovskite films, only electron/hole extraction layer and inorganic perovskite layer (i.e. $TiO_2/CsPbI_3$ and $CsPbI_3/spiro-OMeTAD$) are fabricated via same solution-chemistry processing and the same thickness with the fabricated cell, the PL decay dynamics are modeled via accounting the excitations number and distributions according to the one-dimensional diffusion equation^{S1,S2}.

$$\frac{\partial n(x,t)}{\partial t} = D \frac{\partial^2 n(x,t)}{\partial x^2} - k(t)n(x,t) \quad (1)$$

in which $n(x,t)$ is the number of excitations within a certain thickness of perovskite film, $k(t)$ is the PL decay rate without quenching layer, and D is the diffusion coefficient. The PL decay rate $k(t)$ is

a function of decay time at a certain film depth, which is determined from the slope of each decay time point in the fitting PL decay curve with a stretched exponential decay function. For statistics conveniently, we determine a series of decay rate values at different time by choosing the intensity value and time of two adjacent points from the fitting curve. For the initial value

$$k(t) = \partial(y - y_t) / \partial(x - x_t) \Big|_{t=0} \quad (2)$$

For determining the number of excitations $n(x,t)$, we first describe the initial exciton distribution in the boundary of CsPbI₃ film as

$$n(0, t) = D \partial n(x, t) / \partial x \Big|_{x=0} \quad (3)$$

$$n(z, 0) = n(0) e^{-\alpha z} \quad (4)$$

$$n(z, t) = 0 \quad (5)$$

where z is the perovskite layer thickness, $n(0)$ is the initial exciton density and α is absorption coefficient. Then, plug the individual points of fitting PL decay curves into initial exciton distribution equation by SPSS data statistics software. The average diffusion length L_D is given by $\sqrt{D\tau_e}$, where τ_e is the time taken for the PL to fall to 1/e of its initial intensity in the absence of any quencher. In the process of measurement, the excitation pulse was from the glass substrate side of the samples, pulsed at frequencies between 0.1-1 MHz, with a pulse duration of 117 ps and fluence of ~ 0.03 - $3 \mu\text{J}/\text{cm}^2$. Any deviation from this distribution due to the optical attenuation and reflection of the laser pulse at the perovskite/quencher interface was assumed to be negligible.

5. On page 7, line 189 and page 8, line 209/210, the authors claim that PVP inhibits defect recombination, which is highly speculative based on the data shown. In order to provide evidence for this, the authors could for instance compare to cubic CsPbI₃ without PVP, which can be temporarily maintained in the black phase by heating to $> 300 \text{ C}$ followed by rapid cooling (Ref. 14 in manuscript).

Answer and Modification:

For a sufficient proof, the defect recombination behavior of cubic CsPbI₃ films with and without PVP is represented via steady-state PL spectra, as shown in Fig. S17 of Supplementary Information. The PL spectra of cubic CsPbI₃ films without PVP were measured under temporary black phase by heating to $> 300 \text{ }^\circ\text{C}$ followed by rapid cooling. In the interior of perovskite films, the emergence of grain defects induces both trap-assisted recombination and possible Auger-like recombination. These nonradiative recombination pathways play significant role and appear as photoluminescence (PL) inactive (or dark) areas on perovskite films^{R7}. These enhanced nonradiative recombination pathways result in PL quenching^{R8,R9}. In Fig. S17 of Supplementary Information, compared to cubic CsPbI₃ films with PVP, the PVP-free ones exhibit relatively weak PL intensity. The quenched PL intensity can be attributed to grain surface and internal defects which cause a great deal of nonradiative recombination pathways for PVP-free cubic CsPbI₃ films.

Fig. S17 The steady-state PL spectra of cubic CsPbI₃ perovskite films fabricated with PVP and without PVP.

6. In figure 4b), the initial shape of the red TRPL decay looks a bit odd, like it's off-scale.

Answer:

In fact, there is undecayed curve for α -CsPbI₃ (red TRPL) after 600 ns, just as shown in the following Figure. Due to the significant difference in lifetime between the α -CsPbI₃ (red TRPL) and β -CsPbI₃ (black TRPL), the purpose of cutting off the TRPL curve at 600 ns is for a clear presentation of both curves. The similar presentation of TRPL curves is also reported in several works related to perovskite solar cells^{R5,R10}. For not causing misunderstanding, we supplement the following figure of whole TRPL curve for α -CsPbI₃ in supplementary information.

Fig. S18 Time-resolved photoluminescence (TRPL) spectra of cubic CsPbI₃ films deposited on glass substrates.

Minor issues:

1. The term ‘stability’ is used in different contexts. It should therefore be clear to the reader whether the authors refer to crystal phase stability or thermal/moisture stability. Also, although a crystal phase can be metastable, it does not make sense to refer to a phase transition as metastable (p2 line 51).

Answer and Modification: For clear for readers to understand the “stability” in different context, the ambiguous “stability” has been classified and defined as “phase stability” or “thermal/moisture stability” in manuscript with red font. And the “metastable” in Line 4, Paragraph 2 of Introduction

Section has been removed.

2. Page 6, line 161, please specify the wt-% of PVP used for the optical, electrical and photovoltaic characterization.

Modification:

10 w% of PVP has been supplemented in Line 2, Paragraph 6 of Results and discussion Section in the manuscript with red font.

3. Figure captions S6 and S7 “PVP proportion fixed”. Please specify the concentration.

Modification:

The specific concentration of PVP has been supplemented in the figure caption of S6 and S7 with red font.

4. Page 7, line 207, ‘compared with other inorganic perovskite solar cells’: references are missing.

Modification:

The missing references have been supplemented in Line 13, Paragraph 8 of Results and discussion Section with red font.

Response to Reviewer #3

1. Many typos in the manuscript (for examples, “Sate” Key Program of National Natural Science). The authors should check the manuscript thoroughly.

Modification: The manuscript has been carefully checked thoroughly and the spelling and typos mistakes have been corrected with red font.

2. Comparison of current state-of-the-art Cs-based photovoltaics (both CsPbI₃ and CsPbIBr) should be put in a table.

Modification: The table of “Comparison of current state-of-the-art Cs-based photovoltaics” has been supplemented in Supplementary information as Table S2.

3. 65% FF is still lower than organometallic lead halide perovskites, any suggestion for further improvement?

Answer:

Due to the distinctive crystal structure of inorganic perovskite CsPbX₃, the morphology of CsPbX₃ film is more accidented compared to organometallic lead halide perovskites although both of them have high surface coverage, which might be the primary reason for the low FF. So, firstly, a better film quality with smoother surface morphology and closer connection between the grains can be expected in future by film and solvent engineering^{R11,R12}.

In addition, the recombination of electrons and holes caused by the obstacle in transport and extraction of carriers can be another reason for low FF in perovskite solar cells. So, secondly, the design and application of novel hole transport materials in inorganic perovskite solar cells might improve the FF^{R13}. Thirdly, boosting the hole extraction efficiency by interfacial engineering might be beneficial to the increase of FF^{R14}.

4. The last and most importantly, the authors showed that PVP stabilized CsPbI_xBr_{3-x} film photographs and UV-vis absorption spectra but no solar cells performance (only CsPbI₃ ones). Authors should at least fabricate CsPbI₂Br devices with reported PVP additive and show their performance since, to date, highest reported Cs-based inorganic perovskite solar cell efficiency (PCE = 11.8%) is from the CsPbI₂Br active layer.

Modification:

According to the suggestion, the CsPbI₂Br devices with PVP additive have been fabricated and their photovoltaic performances have been supplemented in Supplementary Information (Fig. S19). Compared to the cubic CsPbI₃ perovskite solar cells, the CsPbI₂Br ones exhibit a slightly decreased J_{sc} of 14.21 mA cm⁻² and an increased V_{oc} of 1.12 V, which might be owing to the widened bandgap originated from Br incorporation. The acquired cubic CsPbI₂Br perovskite solar cells with PVP present a best efficiency of 10.07% and an average efficiency of around 8%.

Fig. S19 The J-V curves for the best cubic CsPbI₃ and CsPbI₂Br cell with PVP measured by forward and reverse scans. Histogram of average efficiencies for 30 devices of cubic CsPbI₃ are summarized (inset).

References

- R1. Zhang, Z. et al. PVP Protective Mechanism of Ultrafine Silver Powder Synthesized by Chemical Reduction Processes. *J. Solid State Chem.* **121**, 105–110 (1996).
- R2. Liu, Z.-Q. et al. ZnCo₂O₄ Quantum Dots Anchored on Nitrogen-Doped Carbon Nanotubes as Reversible Oxygen Reduction/Evolution Electrocatalysts. *Adv. Mater.* **28**, 3777–3784 (2016).
- R3. Yin, Q. et al. Micellization and aggregation properties of sodium perfluoropolyether carboxylate in aqueous solution. *J. Ind. Eng. Chem.* **42**, 63–68 (2016).
- R4. Ahn, N. et al. Thermodynamic regulation of CH₃NH₃PbI₃ crystal growth and its effect on photovoltaic performance of perovskite solar cells. *J. Mater. Chem. A* **3**, 19001–19006 (2015).
- R5. Stranks, S. D. et al. Electron-Hole Diffusion Lengths Exceeding 1 Micrometer in an Organometal Trihalide Perovskite Absorber. *Science* **342**, 341–344 (2013).
- R6. Shaw, P. E. et al. Exciton Diffusion Measurements in Poly(3-hexylthiophene). *Adv. Mater.* **20**, 3516–3520 (2008).
- R7. Li, C. et al. Real-Time Observation of Iodide Ion Migration in Methylammonium Lead Halide Perovskites. *Small* **13**, 1701711 (2017).
- R8. Zhao, L. et al. Electrical Stress Influences the Efficiency of CH₃NH₃PbI₃ Perovskite Light Emitting Devices. *Adv. Mater.* **29**, 1605317 (2017).
- R9. Mamun, A. A. et al. A deconvoluted PL approach to probe the charge carrier dynamics of the grain interior and grain boundary of a perovskite film for perovskite solar cell applications. *Phys. Chem. Chem. Phys.* **19**, 9143–9148 (2017).
- R10. Ma, Q. et al. Hole Transport Layer Free Inorganic CsPbI₂Br₂ Perovskite Solar Cell by Dual Source Thermal Evaporation. *Adv. Energy Mater.* **6**, 1502202 (2016).
- R11. Xie, F. et al. Vertical recrystallization for highly efficient and stable formamidinium-based inverted-structure perovskite solar cells. *Energy Environ. Sci.* **10**, 1942–1949 (2017).
- R12. Chen, H. et al. Solvent Engineering Boosts the Efficiency of Paintable Carbon-Based Perovskite Solar Cells to Beyond 14%. *Adv. Energy Mater.* **6**, 1502087 (2016).
- R13. Arora, N. et al. Perovskite solar cells with CuSCN hole extraction layers yield stabilized efficiencies greater than 20%. *Science* [10.1126/science.aam5655](https://doi.org/10.1126/science.aam5655) (2017).
- R14. Yang, G. et al. Interface engineering in planar perovskite solar cells: energy level alignment, perovskite morphology control and high performance achievement. *J. Mater. Chem. A* **5**, 1658–1666 (2017).

Dear Editor and referees,

Thanks sincerely for your patient treatment of this manuscript again. We have revised the manuscript according to your kind advices and referee's suggestions. We sincerely hope this manuscript will be finally acceptable to be published on Nature Communication soon. Thank you very much for all your help and looking forward to hearing from you soon.

Reviewer #3

The authors revised the manuscript properly. One missing information of stabilized efficiency curve (stabilized output current density and stabilized PCE vs. time when the device is biased at a maximum power point voltage) should be provided. Cs-based perovskite solar cells sometimes show only 70% or even half of stabilized PCE compared with the J-V sweeping PCE. If the authors can provide such data and related discussion in the main-text of the paper, I will strongly support the publication.

Answer and Modification:

According the suggestion, we fabricated new cubic phase CsPbI₃ perovskite solar cells, and selected the best-performance device for the stable photocurrent and output measurements. The Supplementary data are shown in Supplementary Figure 19 in Supplementary Information. The related discussion is shown in Line 10-12, Paragraph 3 in page 7 in Optical and photovoltaic performance Section.

Reviewers' comments:

Reviewer #1 (Remarks to the Author):

All my concerns were solved, and I have no further comments.

Reviewer #2 (Remarks to the Author):

In my opinion, the authors have adequately addressed my concerns and I therefore recommend publication of this revised manuscript.

Reviewer #3 (Remarks to the Author):

The authors revised the manuscript properly. One missing information of stabilized efficiency curve (stabilized output current density and stabilized PCE vs. time when the device is biased at a maximum power point voltage) should be provided. Cs-based perovskite solar cells sometimes show only 70% or even half of stabilized PCE compared with the J-V sweeping PCE. If the authors can provide such data and related discussion in the main-text of the paper, I will strongly support the publication.

REVIEWERS' COMMENTS:

Reviewer #3 (Remarks to the Author):

The authors provided all the needed information. The manuscript is now good for publication as a VIP paper.